# Oocyte Cryopreservation in Patients with Endometriosis: Current Knowledge and Number Needed to Treat

**DOI:** 10.3390/jcm11154559

**Published:** 2022-08-04

**Authors:** Laurie Henry, Julie Vervier, Astrid Boucher, Géraldine Brichant, Olivier Gaspard, Soraya Labied, Carine Munaut, Stéphanie Ravet, Michelle Nisolle

**Affiliations:** 1Center for Reproductive Medicine, University of Liège, Boulevard du 12ème de Ligne 1, 4000 Liege, Belgium; 2Obstetrics and Gynecology Department, University of Liège, Boulevard du 12ème de Ligne 1, 4000 Liege, Belgium; 3Laboratory of Tumor and Development Biology, Giga-Cancer, University of Liège, 4000 Liege, Belgium

**Keywords:** endometriosis, oocyte vitrification, fertility preservation

## Abstract

The rise of oocytes cryopreservation (OOC) in assisted reproductive techniques allows fertility preservation (FP) in an increasing number of indications. Endometriosis, a highly prevalent disease, potentially impairing ovarian reserve, seems, therefore, an interesting indication for it. The purpose of this study is to summarize the available evidence concerning FP by OOC in women with endometriosis and to calculate the number needed to treat (NNT). In total, 272 articles related to this topic were identified in PubMed. Eight studies were eligible for the review. In order to shed some light, a SWOT analysis was performed and the argument pros and cons were developed. The NNT calculated of OOC was 16, meaning that 16 women need to perform an OOC for one of them to have a child that she would not have had without this technique. In conclusion, OOC must be discussed with patients who suffer from endometriosis since it is an effective technique of FP, which can allow these patients to succeed a pregnancy that they otherwise would not have achieved. Nevertheless, it should not be performed in all patients as there is still a lack of robust socio-economic and risk–benefit data.

## 1. Introduction

Endometriosis is a chronic estrogen-dependent disease, afflicting about 10% of women of reproductive age. This pathology is defined by the presence of endometrium-like epithelium and/or stroma outside the endometrium and myometrium, usually with associated inflammatory and fibrotic processes [1]. Symptoms include pelvic pain, dysmenorrhea, dyspareunia and often infertility, which explain its negative impact on life [2].

Nowadays, treatment consists of medical and surgical therapies. Although new therapeutic agents are being evaluated, the only non-surgical treatments currently available, based on painkillers and hormonal treatments, are not able to cure the disease and symptoms quickly recur when the medication is stopped [3]. Surgery is, therefore, the only potentially curative treatment but surgical management should be individualized, especially regarding endometrioma (OMA), since it causes a reduction in ovarian reserve, as measured by a decrease in AMH level [4,5,6] when it is already reduced by the mere presence of an OMA [7]. Endometriosis surgery worsens the phenomenon induced by endometriosis itself as the disease is associated with infertility through various mechanisms, such as chronic inflammation, increased oxidative stress, impaired angiogenesis and cell cycle dysregulation [1,8].

Since several years ago, oocyte cryopreservation (OOC) has been offered in an increasing number of indications, such as ovum donation programs and especially fertility preservation (FP) in women with cancer, in transgender patients or for AGE-banking [9,10]. It is, therefore, legitimate to also propose this technique for patients suffering from endometriosis, since it brings the additional benefit of its relatively low invasiveness and, particularly, its absence of negative impact on ovarian reserve [11]. However, there are few data and recommendations about FP for these patients. Even if the relevance of ovarian testing in guiding FP options or treatment decisions, in endometriosis, patients remains inconclusive; clinicians should be aware that in patients with endometriosis, the involvement of the ovaries and the radicality of surgery influence ovarian reserve [12]. The first practical recommendations of FP in endometriosis patients were published by a French group. They recommend offering FP for bilateral OMA > 3 cm or an OMA on a single ovary. They also suggest doing so before cystectomy if the ovaries are easily accessible for retrieval, in order to increase the number of oocytes cryopreserved [13].

Therefore, fertility counselling in patients with endometriosis is currently a hot topic as most women consider their knowledge on fertility preservation insufficient. Adequate information on fertility and reproductive choices, such as oocyte vitrification, should be incorporated into follow-up visits for endometriosis patients [14].

This review aims to provide an update on current knowledge concerning fertility preservation by oocyte vitrification in patients with endometriosis but also to shed light on this problem from another angle than the one generally approached, by calculating the number needed to treat (NNT) to have one additional live birth. The final purpose is to promote awareness of the importance of fertility counseling among physicians who follow these patients.

## 2. Materials and Methods

### 2.1. Search Strategy

The PubMed database (National Library of Medicine, https://pub-med.ncbi.nlm.nih.gov/) was searched up to 30 November 2021. A combination of Medical Subject Heading (MeSH) descriptors were used: “endometriosis” or “endometrioma” and “oocytes vitrification”, “fertility preservation”, “oocyte vitrification”, “oocyte freezing” or “oocyte cryopreservation”. We also manually searched references for additional relevant publications.

### 2.2. Screening of Publications

After the systematic search using the specific MeSH, two of the authors (LH and JV) performed separately a global screening based on published protocols and method sections from publications and articles recognized as relevant by both were analyzed in more detail. All prospective and retrospective studies published in English were included.

In total, 85 studies were identified upon initial search. Following exclusions and search of additional relevant publication, 8 eligible articles were identified including 1 case report [15], 5 retrospective studies [11,16,17,18,19] and 2 prospective studies [20,21] (Figure 1). These articles are summarized in Table 1.

### 2.3. Calculation of Number Needed to Treat (NNT)

The NNT calculated in this review was based on the number of patients under 35 years old with endometriosis undergoing OOC needed for one additional live birth in this group of patients. Because no randomized controlled trial was available to calculate this number, data published in two studies were used. The first one was the publication of Cobo et al. concerning FP by OOC in patients with endometriosis [11] and the second one was the study published by Somigliana et al. about the use and effectiveness of IVF in patients after conservative surgery for endometriosis without prior OOC [22].

In the publication of Cobo et al. [11], the cumulative live birth rate (CLBR) for women under 35 years of age was 52.8% in patients who underwent surgery before cryopreservation and 72.5% in patients who realized fertility preservation before surgery.

The findings by Cobo et al. [11] suggest that FP in older women would not be as effective whether or not they underwent surgery. In contrast, younger women would be good candidates for FP before surgical treatment because they would require fewer stimulation cycles due to their better ovarian reserve and would have a better reproductive prognosis. In this younger group, the absolute risk reduction (AAR) was calculated (47.2–27.5) and corresponded to 19.7%. This means that performing surgery before cryopreservation on 100 patients, therefore, avoided 19.7 failures. The NNT was consequently 5.076 (1/19.7 × 100). Therefore, for every 5.076 patients with surgery before cryopreservation, one failure was avoided. This seemed, thus, very promising.

Nevertheless, it should be remembered that this cohort included only women who underwent FP. Thus, it did not represent actual effectiveness. It was not possible to exclude that a proportion of those women could have achieved pregnancy spontaneously or with an in vitro fertilization procedure, without previous egg banking. To be more accurate, additional benefit should be calculated instead of live birth rate obtained.

In order to calculate the additional benefit that FP could have on preoperative management of endometriosis, a simulation was performed with a cohort published by Somigliana et al., in 2009 [22] including an unselected cohort of 438 women operated on for endometriosis who wanted a pregnancy. The characteristics examined were the use and effectiveness of IVF. The mean age was 32.4 years old. In this cohort, there were 40% (n = 194) of spontaneous conceptions, 32% (n = 139) of the patients started IVF cycles and 28% (n = 105) did not seek a spontaneous pregnancy but did not begin IVF. Of those 139 patients entering IVF cycles, 35 pregnancies were obtained.

In this cohort of young and operated-on women, one can imagine that there will be an additional benefit of FP compared to IVF with fresh oocytes. Indeed, if they had FP prior to their surgery, the age at vitrification would have been younger and it is assumed that ovarian reserve will be higher as demonstrated in Cobo et al. [11], thus, IVF failure rate should be reduced. We calculated an overall failure rate of conceptions and finally the NNT considering spontaneous conceptions (Figure 2). This statistical analysis was performed with the collaboration of the Biostatistics and research method center (B-STAT) of the University Hospital of Liege, Belgium.

## 3. Ovarian Hyperstimulation Protocol

Currently, there is still no ideal controlled ovarian stimulation (COS) protocol for patients with endometriosis, neither for IVF nor for oocyte vitrification.

The European Society of Human Reproduction and Embryology (ESHRE) guidelines suggested, in 2014, that a down-regulation with a GnRH agonist for at least three months (and up to six months) before an IVF cycle could increase the odds of clinical pregnancy [23]. This recommendation was based on a review published by the Cochrane in 2006 [24] and has been recently superseded by a new one published in 2019 by Georgiou et al. [25]. The latter concluded that the effect of long-term GnRH agonist therapy is uncertain, especially on live birth rate (LBR), mean number of oocytes and mean number of embryos. Therefore, the extended administration of GnRH agonist prior to assisted reproductive techniques (ART) to improve LBR is no longer recommended by ESHRE [26].

In ART, GnRH-antagonist and GnRH-agonist protocols seem to offer the same chances to obtain at least one live birth following utilization of all fresh and frozen embryos after the first ART cycle, as demonstrated by Toftager et al. [27]. In this study, data were stratified according to the primary cause of infertility and this showed that the CLBR was not significantly different between the antagonist group (23%) and the agonist group (15.4%), but the number of patients was low in these subgroups. As already stated by the ESHRE recommendations in 2014, the GnRH antagonist protocol in women with minimal to mild endometriosis and endometrioma could be equivalent to the agonist GnRH one [23]. However, the antagonist protocol has the major advantage to significantly reduce the risk of severe ovarian hyperstimulation syndrome (OHSS) [28]. The latest ESHRE recommendations, published in 2022, state that both GnRH antagonist and agonist protocols can be offered based on patients’ and physicians’ preferences as no difference in pregnancy nor live birth rate has been demonstrated [26].

Patients with endometriosis are often on long-term a combined oral contraceptive pill (COCP) to minimize their symptoms, but COCP pre-treatment is not recommended in the GnRH antagonist protocol because of reduced LBR. However, there is no evidence that COCP alters the number of oocytes retrieved [29].

Before OOC, if data obtained in IVF are extrapolated, both the GnRH-agonist and antagonist protocol can be applied. Nevertheless, in the studies detailed in Table 1, specific to oocyte vitrification for endometriosis, the most frequently used protocol for COS was antagonist. Agonist protocol was only used in four studies: a case report of one patient [15], two retrospective observational studies but with an unknown number of patients in Raad et al. study’s [17], in only 13.1% of patients in the largest cohort of Cobo et al. [11] and, finally, only in 5% of patients in an observational study by Santulli et al. [21]. Unfortunately, no data on the number of mature oocytes vitrified according to the protocol were published.

The progestin-primed ovarian stimulation (PPOS) protocol can also be performed. It requires fewer injections than a conventional protocol and may, thus, be considered more comfortable for the patients. In a study published by Mathieu d’Argent et al., comparing the PPOS and the traditional GnRH-antagonist protocols, there were similar results in terms of number of oocytes vitrified for both groups, but the PPOS protocol was cheaper and, consequently, the medico-economic analysis was in favor of the PPOS protocol. In the PPOS group, different types of progestin were used but there was no statistical analysis between subgroups [20]. This kind of COS was also studied in other research published in 2020 [30]. Three different progestins—medroxyprogesterone acetate, dydrogesterone and progesterone—were combined with hMG (human menopausal gonadotropin) and used in patients with advanced endometriosis, diagnosed and treated by surgery prior to COS. In this study, all three protocols were equivalent in terms of fertilization and pregnancy outcomes, but a higher number of oocytes were retrieved in the medroxyprogesterone acetate group. Therefore, this PPOS protocol seems interesting for endometriosis patients, but more studies are needed to evaluate its effectiveness on larger cohorts and to analyze its impact on pregnancy rate and LBR.

## 4. Ovarian Pick-Up

The technique of oocyte retrieval by trans-vaginal access is the same for OOC or IVF. In case of endometriosis, the procedure of ovarian pick-up (OPU) can be challenging and may affect the individual operator’s performance rate compared to traditional OPU [31]. The ESHRE Working Group on Ultrasound in ART published, in 2019, recommendations about OPU that included endometriosis patients [32].

As endometriosis is a risk factor for pelvic infection, administration of antibiotics is recommended shortly before or during OPU, especially in the case of OMA. Nevertheless, there is no standard protocol recommended for antibiotic prophylaxis. In studies reported in Table 1, only Kim et al. and Santulli et al. specified that antibiotic prophylaxis was provided for all the patients who underwent OPU and no adverse events of infection were described [19,21].

A prospective study published in 2018 comparing technical difficulties of OPU between women with (n = 56) and without (n = 227) OMA(s) showed that OPU in women with ovarian OMAs is trickier. The main complications described were transfixion of the cyst, contamination of the follicular fluid and a more frequently incomplete follicular aspiration. Anyway, the magnitude of these increased difficulties is modest and does not justify systematic surgery before IVF [33]. However, caution is advised during OPU and OMA should not be aspirated nor punctured to prevent contamination of the follicular fluid and to decrease the risk of intra-abdominal infection. Nevertheless, piercing the OMA is often the only way to avoid losing an important number of oocytes. Consequently, patients should be counselled preoperatively and informed consent should be obtained. If an OMA is accidentally punctured, the needle should be immediately withdrawn and flushed with media and the collecting tube should be changed [32]. According to the French recommendations, drainage—with a technique such as sclerotherapy—rather than cystectomy should be performed in first line if OMAs are too bulky and/or if they prevent easy access to the ovaries for retrieval [13].

## 5. Number of Retrieved Oocytes

Among the eight studies highlighted in our review, the number of mature oocytes retrieved and cryopreserved by cycle was in a range between 4.8 [19] and 7.2 [17] (Table 1).

However, the impact of previous surgeries must be considered. Indeed, although OMA surgery is sometimes necessary before considering an OPU, it is known that it has a deleterious effect on the number of oocytes harvested, as shown in the studies included in our review. In the Raad et al. study [17], there were 8.3 ± 5.2 vs. 5.3 ± 3.7 mature oocytes (*p* < 0.01) in patients without previous ovarian surgery when compared with those with a previous OMA excision. Further, there were 10.3 ± 7.8 mature oocytes/patient in patients without surgery vs. 8.5 ± 4.8 in patients with unilateral surgery vs. 8.0 ± 5.7 in patients with bilateral surgery in the Cobo et al. study [11]. In a cohort published by Santulli et al., there were 11.7 ± 6.6 mature oocytes harvested in patients without surgical history and 8.6 ± 6.0 in patients with previous OMA surgery [21]. Those results were confirmed in the study of Mathieu d’Argent et al. (*p* = 0.035) [20].

Although the effect of prior surgery is clearly demonstrated, the presence of deep endometriosis (versus superficial endometriosis alone), the location of endometriosis, the presence of endometrioma during the stimulation and the size of endometriomas were not associated with the number of retrieved oocytes (*p* > 0.05). Nevertheless, the meta-analysis, conducted by Muzii et al., concluded that ovarian reserve, evaluated by AMH level, is significantly reduced in patients with OMA compared to both patients with other benign ovarian cysts and patients with healthy ovaries [7], which is detrimental to the number of oocytes that can be expected following COS. However, a recent retrospective study of 50 IVF cycles in patients (<37 years old) with severe endometriosis showed that the low value of AMH did not affect oocyte quality and pregnancy outcome in IVF patients [34].

## 6. Return and Success Rates

There are, unfortunately, few data in terms of return rate, pregnancy (PR) or LBR after OOC for endometriosis. Among the eight selected studies in our review, only three mentioned these results.

In the study of Kuroda et al., embryos were cryopreserved, not oocytes, which implies a more accomplished reflection in terms of pregnancy wish, since it is already a couple project. However, there were no data concerning the return rate but 37.5% patients (6/16) with OMA experienced live birth in this study [18].

The return rate of patients with vitrified oocytes varies between 13% [16] and 46.5% [11]. In this latest publication from Cobo et al., which is the one with the highest patient number (n = 485), 225 babies were born with a CLBR/patient of 46.4%. CLBR was higher (*p* < 0.05) in the subgroup of patients of ≤35 years of age and without previous ovarian surgery (72.5%). In this study, a comparison was performed between patients with endometriosis vs. elective fertility preservation (EFP) patients used as a historical control group. The oocyte survival rate, implantation rate, pregnancy rate and CLBR were higher in the young (≤35 years old) elective FP patients compared with endometriosis matching-age patients (*p* < 0.05). These data were processed by the authors to calculate the CLBR according to the number of frozen oocytes. They concluded that CLBR increased as the number of oocytes used per patient rose, reaching 89.5% using 22 oocytes and even 95.4% using approximately 20 oocytes in ≤35-year-old women, without statistical differences between EFP and endometriosis patients [35]. Nevertheless, obtaining 20 vitrified oocytes can be easy in young women but harder in the case of endometriosis where the ovarian reserve can be altered, especially after previous surgery.

These data corroborate the existing literature on the impact of endometriosis on IVF. This poorer prognosis of patients with endometriosis is mainly related to the oocyte rather than the endometrial receptivity [36,37,38]. The systematic review and meta-analysis performed by Horton et al. demonstrated a negative impact of the pathology on various IVF parameters, such as a reduction in mature oocytes in the more severe subtype and those affected by endometrioma [39]. Although endometriosis does not have a negative impact on oocyte morphology in IVF-ICSI [40], it alters oocyte quality, as reflected, among other things, by a differential transcriptomic profile [41] and a lower cytoplasmic mitochondrial content [42]. However, the lower results observed do not seem to be related to an increase in the aneuploidy rate [11,43].

## 7. Number Needed to Treat

The NNT is the number of people who need to be treated during a specific period to cure or prevent an additional case of the disease under consideration. It is a simple tool that can be used in daily practice to explain statistics as well as the clinical relevance of new treatments. Considering fertility and especially the usefulness of OOC prior to endometriosis surgery, the NNT corresponds to the number of women in whom cryopreservation before surgical treatment of endometriosis must be performed to guarantee one supplemental live birth.

Considering the success rates of OOC, depending on whether it was performed before or after surgery, according to the data published in the study of Cobo et al., we were able to estimate a failure rate for the 139 patients who performed IVF in the study of Somigliani et al. If the OOC was performed before surgery, the failure rate is 38.22%, while it was 65.61% if there was no OOC.

The overall failure rate of conception was, therefore, calculated for both groups considering the whole population included in the cohort of Somigliana et al. and was 8.73% in the OOC group compared to 14.98% in the group of patients without OOC. These numbers allowed us to calculate the absolute risk reduction, which was 6.25, and, therefore, the NNT. This NNT was 16 (Figure 2), a far more realistic number and less biased than the 5 calculated previously, as described in Materials and Methods.

## 8. Impact of Oocyte Preservation on Pain and Endometriosis Recurrence

The onset or exacerbation of endometriosis-related pain after ovarian stimulation for OOC, which is sometimes repetitive, is only lightly discussed in the cited articles. In the study of Kuroda et al., there was no comment about pain recurrence but a surgery was planned in all patients shortly after the OOC [18]. Kim et al. mentioned that there was no increase in the size of OMA after COS but, again, surgery was scheduled soon after the COS [19]. Santulli et al. only mentioned that painful symptoms during ovarian stimulation did not increase [21]. Therefore, considering the short duration of ovarian stimulation and the fact that surgery is quite often planned shortly after ovarian stimulation, the probability of disease progression seems to be low.

This is confirmed in the systematic review from Somigliana et al., which highlighted that IVF does not worsen endometriosis-related pain symptoms (moderate-quality evidence), does not increase the risk of endometriosis recurrence (moderate-quality evidence) and that the impact of IVF on OMA, if present at all, is mild (low-quality evidence). It was also mentioned that deep invasive endometriosis might progress with COS but with very-low-quality evidence [44].

## 9. Discussion

Oocyte vitrification is a relatively new ART technic, of which indications only grow with its widespread use. It is, therefore, natural that a pathology like endometriosis, which could alter ovarian reserve and fertility, became an indication of this fertility preservation method. Unfortunately, there is currently no randomized study published about OOC in endometriosis patients. Among the eight studies highlighted in this review, only two are prospective cohort studies; the others are retrospective data or even a case report. This small number of cohort studies is a limitation to this review, but it demonstrates that OOC for endometriosis patients is an emerging field in ART.

In this study, the success rate of oocyte vitrification was approached under a new angle, less used in fertility but well known in medicine, the NNT. Through our concept, we were able to calculate an NNT of 16. This means that 16 patients under 35 years old must undergo OOC before endometriosis surgery, to allow one of them to have a child later on, which she could not have had if the OOC had not been performed. This is important information to mention to women when explaining the impact of endometriosis on their fertility and the possibility of performing an OOC. Nevertheless, the fact that two cohorts of patients had to be used to obtain this result is a limitation in our study. In order to be able to calculate a correct NNT, prospective studies should be performed with a cohort, including women who did not use oocyte cryopreservation.

Given this paucity of information regarding OOC in endometriosis patients and to provide a comprehensive view, we supplemented these data with what is known about the specificities of IVF in patients with endometriosis. This allowed us to draw up an overview of this technique and to highlight its Strengths, Weaknesses, Opportunities and Threats, which are summarized in a SWOT analysis, updated from the one published in 2018 by Streuli et al. [45] (Figure 3). We detailed, thereafter, the pros and cons in oocyte cryopreservation in endometriotic patients.

The arguments in favor of FP in endometriosis patients are as follows.

As discussed previously in this review, endometriosis as such can alter the ovarian reserve, both quantitatively and qualitatively [1,7,8,36,37,38,39,42,46].

Ovarian surgery for OMA, even performed by expert surgeons, affects ovarian reserve [4,5,6,7,47]. Since presence of endometrioma is found in 17–44% of patients with endometriosis, surgical treatment of endometriosis often implies ovarian surgery [47].

Endometriosis is a recurrent disease and recurrence occurs in 40–50% at 5 years [48], even if it has been recently showed that a post-operative hormonal suppression could reduce disease recurrence and pain [49]. Multiple surgeries for endometriosis are known to have no positive impact on fertility and recurrence of ovarian surgery will further worsen the reproductive prognosis of the patient [23].

Medical treatments currently proposed have little or no impact on fertility. Hormonal treatments are contraceptive and analgesic treatments have limited efficacy over time. Overall, approximately 50% of women with endometriosis have recurrent symptoms over a 5-year period, regardless of treatment approach [50].

Age and ovarian reserve (number of vitrified oocytes) are primary factors for a successful fertility preservation. In recent years, a tendency to delay first pregnancy has been observed worldwide (29.3 years in Europe [51]), so we could assume that younger women (and/or single women) with endometriosis would not necessarily want a pregnancy in the short to medium term. When diagnosed with endometriosis, women could not be ready to have a child. Cryopreserving their gametes at a young age before surgery would allow a better quality of banked oocytes and give them more time to think about their reproductive desires.

OOC is a validated technique in fertility preservation in other indications, such as oncofertility. It has been reported as the best option because of its low negative impact on ovarian reserve and low associated morbidity compared to ovarian tissue cryopreservation, the other FP technique available [9]. Furthermore, the CLBR in patients who have used this technique is relatively high [11] and the NNT is 16, which is quite interesting.

Finally, the return rate after oocyte vitrification in endometriosis patients seems high, reaching 46.5% in the study of Cobo et al. [11], suggesting that fertility preservation in patients with endometriosis is important for their reproductive future.

The arguments against FP in this indication are rather socio-economic or related to the type of endometriosis.

When it does not involve an ovarian procedure, surgical treatment of endometriosis is certainly indicated in women with pelvic pain and it allows couples to conceive, often without IVF [52]. The ESHRE guidelines recommend that in symptomatic patients, clinicians should perform operative laparoscopy rather than diagnostic laparoscopy alone, in order to increase the ongoing pregnancy rate in patients with endometriosis at stage I/II [26]. Nevertheless, if the first surgery is not followed by a pregnancy, a second operation for endometriosis is not effective on infertility and should not be recommended [53].

Endometriosis is a relatively common disease and oocyte cryopreservation is expensive and exposes women to some clinical risks. Therefore, the systematic inclusion of women with endometriosis in a fertility preservation program would have considerable clinical, logistical and financial impact [54], with still many questions remaining unanswered. There are currently insufficient data to support fertility preservation for all women with endometriosis [55].

## 10. Conclusions

This review provides an update on the current knowledge concerning fertility preservation by oocyte vitrification in patients with endometriosis, which is an emerging field of medical reproduction. Currently, the indication must be carefully considered and FP should not be offered to all patients suffering from endometriosis. Indeed, prospective studies on large cohorts, especially cost-effectiveness and risk–benefit studies, must still be carried out in order to better define the indications for this technique.

Nevertheless, clinicians must address the sensitive topic of fertility in patients with endometriosis, and even if the true benefit of OOC remains unknown, they should discuss the pros and cons of fertility preservation. Patients must have all the information to decide, in full knowledge of the facts, whether or not to perform an OOC.

## Figures and Tables

**Figure 1 jcm-11-04559-f001:**
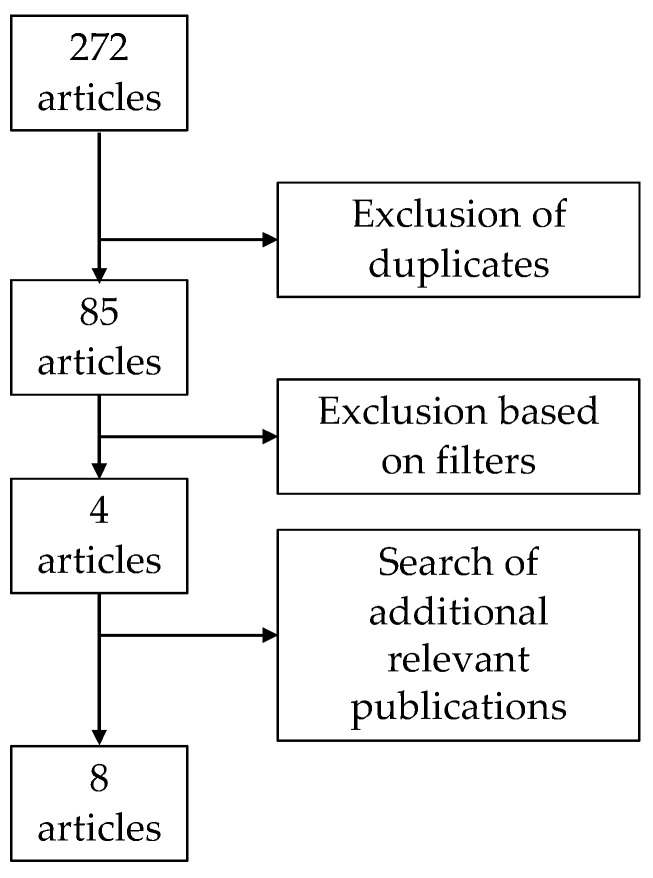
Flow diagram for the selection of the articles included in this review.

**Figure 2 jcm-11-04559-f002:**
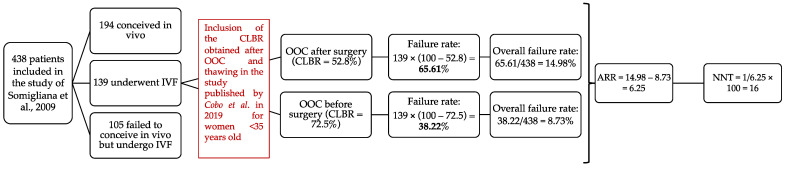
Calculation of number of women in whom cryopreservation before surgical treatment of endometriosis must be performed to guarantee one live birth. IVF: in vitro fertilization, ARR: absolute risk reduction, NNT: number needed to treat, CLBR: cumulative live birth rate. Success rate = patients undergoing IVF in the publication of Somigliana et al. [22] × CLBR [11]. Failure rate = patients undergoing IVF in the publication of Somigliana et al. [22] × (100 – CLBR [11]). Overall failure rate = failure rate/total of patients included in the study of Somigliana [22]. ARR for cryopreservation before surgery was calculated by establishing the difference in absolute risk between the group of patients who underwent surgery before cryopreservation and the second group of patients who realized fertility preservation before surgery. NNT was calculated by 1/ARR × 100. It corresponded to the number of women in whom cryopreservation before surgical treatment of endometriosis must be performed to guarantee one live birth.

**Figure 3 jcm-11-04559-f003:**
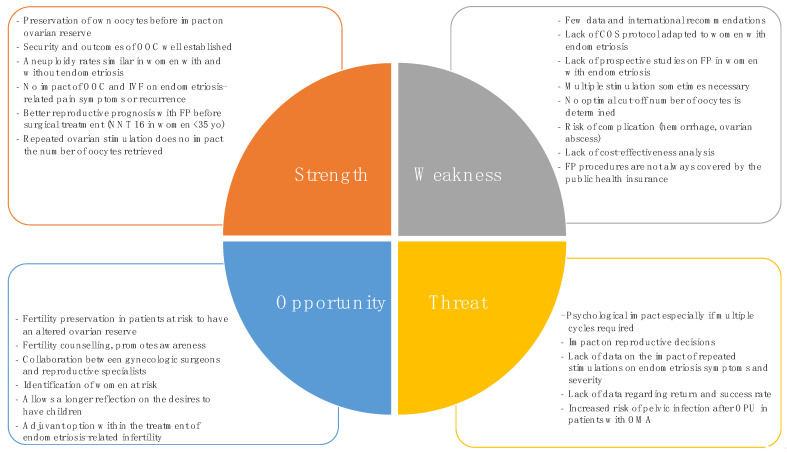
SWOT analysis of oocyte cryopreservation in women with endometriosis. Adapted from I. Streuli et al. [45]. OOC: oocyte cryopreservation, FP: fertility preservation, NNT: number needed to treat, yo: years old, COS: controlled ovarian stimulation, POF: premature ovarian failure, AMH: anti-Mullerian hormone.

**Table 1 jcm-11-04559-t001:** Characteristics and outcomes of the studies included.

References	Study Design	Number of Patients	Mean Age (years) ± SD	Aim	Surgical History for Endometriosis	Type of Preservation	Inclusion Criteria	Technique	Duration of Banking	Outcomes: Mean Number of Matures Oocytes Cryopreserved	Outcomes: PR	Limitations
Elizur et al., 2009 [15]	Case report	1	25	First report of FP with OOC in an endometriosis patient	Right salpingo-oophorectomy; 2 procedures for endometriosis-related adhesions	OOC	Nulliparous woman with severe endometriosis, heavy surgical history, and low OR	3 cycles of COS and ovarian pick-up: 2 with mid-luteal GnRH agonist & 1 with GnRH antagonist-protocol	NR	21	NR	Case report of a single case; No data about pregnancy outcomes
Garcia-Velasco et al., 2013 [16]	Retrospective observational study	38	Unknown for the endometriosis subgroup	To evaluate the results of COS for oocyte vitrification in FP for medical and nonmedical indications (including endometriosis)	NR	OOC	560 nononcological patients (38 for endometriosis) and 475 oncological patients	COS with antagonist protocol (with recombinant FSH and/or highly purified hMG)	NR	Not possible to extract data for endometriosis patients	5/38 (13%) patients returned to use frozen oocytes, but PR is unknown	No data specific for patients with endometriosis about the oocyte quality or pregnancy outcomes after fertilization; The type of endometriosis was not described; Retrospective study
Raad et al., 2018 [17]	Retrospective observational study	49	33.9 ± 4.5	To evaluate the results of COS for oocyte vitrification in FP for endometriosis and to evaluate the impact of a previous surgery for OMA on the results	39% of cycles have a previous surgery for endometrioma	OOC	49 patients with a total of 70 COS with punction. 2 patients (4.1%) had superficial endometriosis, 22 (44.9%) had deep infiltrated endometriosis and 35 (71.4%) had OMA. 10 patients were included in both OMA and deep infiltrated phenotype groups.	COS with GnRH antagonist or long agonist protocol (with recombinant FSH)	NR	(1) 7.2 ± 4.9 mature oocytes/cycle (2) 8.3 ± 5.2 vs. 5.3 ± 3.7 mature oocytes (*p* < 0.01) in patients without previous ovarian surgery when compared with those with a previous OMA excision	NR	No data about pregnancy outcomes after fertilization; no control group; Retrospective study
Kuroda et al., 2019 [18]	Retrospective cohort study	16	Unknown for the endometriosis subgroup	To analyze the clinical outcomes and the predictive factors for the therapeutic effect of preoperative embryo cryopreservation combined with endoscopic surgery in infertile women	NR	EC	38 patients with diminished OR, with uterine fibroids and/or OMA, among those 16 had OMA	COS with a clomiphene-recombinant FSH or -hMG cycle or a GnRH antagonist cycle.	NR	NA	6/16 (37.5%) patients with OMA experienced live birth	Pregnancy outcomes were not compared in patients who underwent IVF treatment, surgery only, or no treatment; Small number of patients; No data specific for endometriosis patients; Retrospective study
Cobo et al., 2020 [11]	Retrospective observational study	485	35.7 ± 3.7	To describe the outcome of FP using vitrified oocytes in patients with endometriosis and to determine the impact of ovarian surgery	47.8% of patients underwent OMA surgery before FP (34.9% bilateral surgery, 65.1 % unilateral surgery)	OOC	49 patients with a total of 70 COS with punction. 2 patients (4.1%) had superficial endometriosis, 22 (44.9%) had deep infiltrated endometriosis and 35 (71.4%) had OMA. 10 patients were included in both OMA and deep infiltrated phenotype groups.	COS with GnRH antagonist or agonist protocol	1.7 years (±0.4)	(1) 5.5 ± 5.2 mature oocytes/cycle and 9.4 ± 6.7 mature oocytes/patient (2) 10.3 ± 7.8 mature oocytes/patient in patient without surgery vs. 8.5 ± 4.8 in patients with unilateral surgery vs. 8.0 ± 5.7 in patients with bilateral surgery	Return rate of 46.5%; 225 babies were born: CLBR/patient 46.4% but higher (*p* < 0.05) in the subgroup of patients of ≤35 years of age and without surgery (72.5%)	Low number of cases at stages I–II (2.3%); no high-quality control group (a historical one); Retrospective study
Kim et al., 2020 [19]	Retrospective observational study	34	30.7 ± 5.9	To evaluate the clinical usefulness of OOC for FP in women with ovarian endometriosis before a planned ovarian cystectomy	32% of patients had previous ovarian surgery before COS	OOC	Women diagnosed with ovarian endometriosis on imaging; women for whom ovarian cystectomy was planned owing to the severity of symptoms or increasing size of the endometrioma; and women who underwent OOC before ovarian surgery for fertility preservation	COS with GnRH antagonist protocol and recombinant FSH	NR	(1) 4.8 ± 3.2 mature oocytes/patient (2) 4.1 ± 2.9 in patients with bilateral OMA vs. 5.7 ± 3.4 in patients with unilateral OMA	NR	No data about warming and pregnancy outcomes after fertilization; Small number of patients; The effect of ovarian surgery before FP was not studied; Retrospective study
Mathieu d’Argent et al., 2020 [20]	Prospective cohort study	108	30.3 ± 4.3	To describe FP outcomes in women with endometriosis and to compare an antagonist protocol with a PPOS protocol	27.8% of patients had prior ovarian surgery: 21.5% for OMA and 20.8% for endometriosis	OOC	Women under 40 years-old with endometriosis (OMA +/- deep endometriosis), and alteration of OR	1 cycle of COS with PPOS and antagonist protocols	NR	6.4 ± 5.6 mature oocytes/patient: no statistical difference between PPOS and antagonist protocol; Prior ovarian surgery was associated with the number of retrieved oocytes	NR	No data about the effect of previous ovarian surgery on FP; No data about pregnancy outcomes after fertilization; Prospective study but not a randomized-controlled trial
Santulli et al., 2021 [21]	Prospective observational cohort study	146	31.5 ± 4.4	To determine prognostic factors related to high oocyte yield in FP for women affected by endometriosis	36.3% of patient had previously undergone surgery for endometriosis	OOC and EC	Women who had previously undergone ovarian stimulation for oocyte or embryo vitrification; with a phenotyped endometriosis after imaging (40 women with and 106 without previous surgery); and aged 38 years or younger	COS with long or short GnRH agonist or antagonist protocol	NR	(1) 10.9 ± 6.6 mature oocytes/patient and 6.7 ± 5.1 mature oocytes/patient after the first ovarian stimulation cycle (2) 11.7 ± 6.6 mature oocytes/patient without surgical history and 8.6 ± 6.0 in patients with previous OMA surgery	NR	No data about warming and pregnancy outcomes after fertilization

SD: Standard Deviation; PR: pregnancy rate; FP: fertility preservation; OOC: oocyte cryopreservation; OR: ovarian reserve; COS: controlled ovarian stimulation; GnRH: gonadotropin-releasing hormone; NR: not reported; FSH: follicle-stimulating hormone; hMG: human menopausal gonadotropin; OMA: endometrioma; EC: embryo cryopreservation; NA: not applicable; IVF: in vitro fertilization; PPOS: progestin-primed ovarian stimulation.

## Data Availability

Not applicable.

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
