# Peer review of "Oocyte Cryopreservation in Patients with Endometriosis: Current Knowledge and Number Needed to Treat"

_jcm, 2022, doi:10.3390/jcm11154559_

Round 1

Reviewer 1 Report

1. Minor proofreading errors remain: in vitro should be written in italic; lines: 92, 115, 172, 228, 255, 257, 260, 299, 327, 356.

2. Strong proofreading of Table 1 is required

3. Should be unify: p>0.0; P>0.0 or p>.0

4. Very difficult to read figure 1, it is impossible to read the SWOT analysis figure.

Author Response

First of all, the authors thank the reviewer to have accepted to evaluate this work and for the constructive comments proposed.

  • Proofreading errors have been corrected.
  • Proofreading of Table 1 has been done but may be further improved with the editing department of the journal. I inserted it as an image to improve the layout.
  • P-values have been unify (p>0.0…)
  • The 3 figures have been reinserted but in pdf format to improve the quality
  • The manuscript has been revised for English.

Reviewer 2 Report

I like the topic and I find your results interesting and useful for clinicians that treat endometriosis. I encourage you to make a prospective study to better define the indications for oocyte cryopreservation in patients with endometriosis.

Author Response

Authors thank the reviewer to have accepted to evaluate this work and for the constructive comment proposed.

Reviewer 3 Report

I read with great interest the manuscript, which falls within the aim of this Journal. In my honest opinion, the topic is interesting enough to attract the readers’ attention.My only suggestion is to make the tables more readable and organise them.

Author Response

Authors thank the reviewer to have accepted to evaluate this work and for the constructive comment proposed. Proofreading of Table 1 has been done but may be further improved with the editing department of the journal. The 3 figures have been reinserted but in pdf format to improve the quality.